# Off-Label Benralizumab in Severe Non-Necrotizing Eosinophilic Vasculitis following Critical COVID-19 Disease and in DRESS

**DOI:** 10.3390/jcm11226642

**Published:** 2022-11-09

**Authors:** Irena Pintea, Ioana Adriana Muntean, Carmen Teodora Dobrican, Nicolae Miron, Diana Deleanu

**Affiliations:** 1Department of Allergology and Immunology, Iuliu Hatieganu University of Medicine and Pharmacy, 400012 Cluj-Napoca, Romania; 2Allergology Department, Professor Doctor Octavian Fodor Regional Institute of Gastroenterology and Hepatology, 400162 Cluj-Napoca, Romania; 3Department of Clinical Immunology, Sahlgrenska University Hospital, 41346 Gothenburg, Sweden

**Keywords:** Benralizumab, vasculitis, DRESS, COVID-19, eosinophils

## Abstract

Benralizumab is a humanized recombinant mAb that binds to the interleukin 5 receptor (IL-5R) expressed on eosinophils and is approved for the treatment of severe eosinophilic asthma. There are a series of severe eosinophilic disorders that may benefit from this treatment, and it could be a life-saving therapy. In this paper, we present two severe patients with eosinophil-induced diseases that had a good resolution after one dose of Benralizumab 30 mg. The first case is a severe non-necrotizing eosinophilic vasculitis following critical COVID-19 disease and the second case is a DRESS (Drug Rash with Eosinophilia and Systemic Symptoms Syndrome) due to allopurinol. Conclusions: The successful administration of Benralizumab in rare or severe eosinophilic disease could be an option for life-saving therapies when conventional treatments fail.

## 1. Introduction

Recent years have brought advances in the therapeutic approach of immune-mediated inflammatory diseases, in many cases due to the employment of biologic agents, particularly monoclonal antibodies (mAbs), directed to key constituents of the inflammatory response. Benralizumab is a humanized recombinant mAb of the isotype IgG1k immunoglobulin that specifically binds to the alpha chain of the interleukin 5 receptor (IL-5R) expressed on eosinophils and basophils [1]. It blocks signal transduction via inhibition of the binding of IL-5 and the hetero-oligomerization of the alpha and beta subunits of the IL-5R. In addition, it is an afucosylated IgG which gives it a high affinity for the FcγRIIIα receptor on natural killer cells, macrophages, and neutrophils [2]. Given its tropism to the alpha-subunit of the IL-5R, Benralizumab induces the direct and substantial depletion of circulating peripheral blood eosinophils within the first 24 h post-administration, a biological effect that persists for at least 2 to 3 months [2,3].

Approved by the FDA in 2017, Benralizumab is indicated as a maintenance treatment for patients 12 years or older with severe asthma and an eosinophilic phenotype [4,5,6]. Benralizumab is administered subcutaneously at a dose of 30 mg every 4 weeks for 3 doses, followed by every 8 weeks thereafter. Overall, it is well tolerated with a favorable safety profile. Recent studies tested the putative efficacy of Benralizumab in other diseases, such as PDGFRA-negative hypereosinophilic syndrome [7], chronic obstructive pulmonary disease and peripheral eosinophilia [8], severe chronic rhinosinusitis with nasal polyps [9], chronic spontaneous urticaria [10,11], as well as in rare eosinophil mediated diseases (eosinophilic cystitis, fasciitis, pancreatitis, and cholangitis, Kimura’s disease, eosinophilic cellulitis, eosinophilic granulomatosis with polyangiitis, eosinophilic pneumonia, and hypereosinophilic syndrome) [12,13,14,15].

In this paper, we report on the off-label use of Benralizumab in two difficult-to-treat cases, one with severe non-necrotizing vasculitis following critical COVID-19 disease and another case with severe DRESS (Drug Rash with Eosinophilia and Systemic Symptoms Syndrome) due to allopurinol.

## 2. Compassionate Use of a Single Dose of Off-Label Benralizumab in Severe Non-Necrotizing Vasculitis following Critical COVID-19 Disease

We describe the case of a 46-year-old female patient with critical COVID-19 disease (bilateral pneumonia with damage to more than 60% of the lung, acute respiratory insufficiency, orotracheal intubation, and mechanical ventilation for six days), post-COVID bilateral inferior lobe necrotizing pneumonia, thrombophilia (heterozygous factor V Leiden mutation and heterozygous MTHFR 1298C mutation), thrombosis of the left internal jugular, subclavian, and brachiocephalic vein, gluteal pressure ulcers infected with enterococcus *spp*, pseudomonas *spp*, klebsiella pneumoniae, and acinetobacter baumannii, urinary tract infection with candida glabrata, pseudomonas *spp*, klebsiella pneumonia, and who developed rapidly progressing skin vasculitis, despite immunosuppression with high doses of corticotherapy, affecting more than 60% of the body surface area (Figure 1 and Figure 2) on the 31st day of admission, and associated mild peripheral eosinophilia. The purpuric rash was initially localized over the hands and legs and consisted of round, 1–3 mm in diameter skin lesions with a rapid tendency to coalesce and form plaques, up to the point that within the following 12 h after onset, it was affecting more than 60% of the BSA. Given the rapid progression of the skin rash along with the worsening of the clinical status, despite broad-spectrum intravenous antibiotics, pulse therapy with methylprednisolone was initiated (intravenous injection, 250 mg/day^−1^ for 3 days, with day 31 of admission being the first day of corticosteroid administration). Skin biopsy, which was done on day 32 of admission and the second day of pulse therapy, revealed non-necrotizing small vessel vasculitis with lymphoplasmacytic infiltrates, and rare eosinophils. In addition to the rapid progression of vasculitic lesions, the patient developed slight peripheral eosinophilia despite methylprednisolone administration. Complete clinical control with remission of skin lesions as well as of eosinophilia was obtained with the compassionate use of a single dose of Benralizumab (Table 1). The laboratory work-up included ANA, cANCA, pANCA, and anti-phospholipid antibodies, which brought negative results. In addition to Benralizumab, the patient received a high-dose of intravenous immunoglobulin of 400 mg/kg/day for 5 days for immunosuppressive purposes. Remission of the vasculitis signs was obtained within the first week and, as expected, the complete depletion of eosinophils was documented in the first 24 h after administration of Benralizumab.

## 3. Off-Label Benralizumab in DRESS to Allopurinol

A 78-year-old female patient was admitted to the hospital with a 2-week history of rash and fever. Three weeks before admission, she had taken allopurinol for gout. Prior to her admission, the patient was treated by her family doctor with antihistamines (four tablets/daily) and corticosteroids (40 mg of prednisone equivalent/day for three days, followed by a 5-day taper), without any clinical improvement. At the time of the admission, the patient’s body temperature was 39.2 °C. Physical examination revealed a generalized erythematous skin rash with lamellar desquamation, especially on the arms, thighs, and anterior thorax (Figure 3, Figure 4 and Figure 5), and generalized lymphadenopathy. Peak laboratory values during hospitalization included peripheral eosinophilia of 21.6% (reference range, 1–4) and 2.62# (reference range, 0.05–0.4), altered renal function (blood urea nitrogen test of 183, reference range 18–48, and a serum creatinine level of 1.66, reference range of 0.51–0.95) with consistent normal liver tests. The antinuclear antibody test and urinalysis, as well as chest radiograph, echocardiogram, and ECG, were normal. Viral hepatitis markers (HAV, HBV, and HCV), and testing for mycoplasma and chlamydia, as well as blood cultures, brought negative results. 

The European Registry of Severe Cutaneous Adverse Reactions (RegiSCAR) [16] (Table 2) score was 7. It indicated a definitive diagnosis of drug reaction with eosinophilia and systemic symptoms (DRESS) related to treatment with allopurinol: generalized exanthema, with edema, infiltration, and scaling, fever, lymphadenomegaly, peripheral hypereosinophilia, abnormal renal function, and tests to rule out other possible causes. Treatment with emollients and oral systemic glucocorticosteroids (a high dose of 1 mg/kc per day of prednisone) did not improve the outcome of the disease, which prompted a compassionate use of a single dose of Benralizumab, 5 days after admission. Benralizumab administration brought a significant clinical improvement, with complete resolution of the skin rash within the following 10 days. Concomitantly, lab tests were within normal ranges on the last day of hospitalization, except for peripheral eosinophils, which was 0 the day after Benralizumab injection (Table 3). Since DRESS can relapse after treatment discontinuation, we discussed the necessity of long-term corticotherapy with the patient. A gradual dose reduction in the prednisone regimen was agreed upon and given over 5 weeks (60 mg/d × 5 days, 40 mg/d × 5 days, 20 mg/d × 5 days, 10 mg/d × 5 days, 5 mg/d × 5 days, followed by 2.5 mg/d × 5 days, and then 2.5 mg/d on alternate days for 6 days) to prevent the rapid reconstitution of specific immune responses. The evolution of the case was favorable, both clinically and in terms of lab test results. DRESS did not relapse after the effects of the Benralizumab wore off, more likely due to the benefits of long-term corticosteroid administration. 

In both patients, the blood samples were obtained à jeun, and the complete blood count including eosinophils was analyzed by the SYSMEX-XN-1000 analyzer, while CRP and d-dimers (the latter were abnormal in the first case, following COVID-19 disease and not in the patient with DRESS) tests were performed with the COBAS PRO C 503/E 801 analyzer.

## 4. Discussions

Eosinophils play a crucial role as effector immune cells committed to host defense and as modulators of innate and adaptive immune responses. In addition, they are involved in tissue repair and the immune homeostasis of several non-immunocompetent tissues. The complex network of eosinophils, T helper 2 lymphocytes, B cells, and mast cells, as well as circulating platelets and cells residing at sites of inflammation, mediate host protection against infections. Eosinophils are involved in the immune response in a wide range of diseases, including infectious diseases (parasitic, bacterial, viral, and fungal infections), hematological disorders and cancer (clonal disorders and solid tumors), graft-versus-host disease, as well as immune-mediated diseases (atopic dermatitis, Gleich syndrome, drug hypersensitivity reactions with eosinophilia, eosinophilic gastroenteropathy, and eosinophil-related respiratory diseases) [17].

Eosinophils play an important role in protecting the host against viral infections, and several publications discuss the eosinophils/IL-5 axis in the COVID-19 illness. SARS-CoV-2 infection stimulates a complex activation of the immune response. In the first phase of the infection, the participation of eosinophils is probably appropriate and beneficial, limiting the viral replication, while in the second phase of clinical pulmonary symptoms and the third phase of the immunopathological response, which results in a cytokine storm, they may take part in a maladaptive immune response and immunopathology. Mounting evidence shows that COVID-19 exhibits peculiar features in terms of the immune response [18]. Eosinopenia has been indicated for the prediction of poor prognosis in COVID-19 [19]. The increase in eosinophils over the course of the disease from initially low levels could be a positive indicator of clinical improvement [20]. In contrast, it has been indicated that current studies do not support the use of eosinopenia for the diagnosis of COVID-19 [21]. It has also been suggested that eosinopenia may not be associated with an unfavorable progression of COVID-19 [22] and that eosinopenia may show a more diagnostic—and, eventually, prognostic—value rather than participation in COVID-19’s pathology [23]. Therefore, the role of eosinophils in COVID-19 remains controversial. A study published in 2021 on the behavior of eosinophil counts in recovered and deceased COVID-19 patients over the course of the disease shows that the increase in eosinophils in the recovered patients protected against the exacerbation of inflammation, protection that was lacking in the deceased patients [19].

In recent years, innovative drugs targeting specific cytokines in the eosinophil signaling network have been introduced, such as anti-IL-5 monoclonal antibodies (mepolizumab and reslizumab) and anti-IL5-R alpha chain antibodies (Benralizumab). Another agent, TPI ASM8, a small oligonucleotide designed for inhaled administration, which exploits RNA interference to dampen the expression of CCR3 and the shared IL3-R, GM-CSF-R, and IL5-R beta chain, appears to control eosinophil inflammation, as shown by clinical studies [24]. These agents have been designed to dampen the effects of eosinophilia on target organs, rather than causing a general immune suppression. 

Although a wide range of diseases have been proven to have an eosinophilic substrate, targeted agents are approved for a limited number of them, which in some cases leaves a gap in the treatment arsenal. 

Recent studies tested the putative efficacy of these agents in diseases other than those with FDA approval for subjacent eosinophilic inflammation, particularly in cases with severe evolution that are unresponsive to conventional treatment. 

Benralizumab is currently FDA-approved for severe eosinophilic asthma. However, this agent proved efficacious in other diseases, namely in PDGFRA-negative hypereosinophilic syndrome [7] chronic obstructive pulmonary disease and peripheral eosinophilia [8], severe chronic rhinosinusitis with nasal polyps [9], chronic spontaneous urticaria [10,11], as well as in rare eosinophil mediated diseases (eosinophilic cystitis, fasciitis, pancreatitis and cholangitis, Kimura’s disease, eosinophilic cellulitis, eosinophilic granulomatosis with polyangiitis, eosinophilic pneumonia, and hyper-eosinophilic syndrome) [12,13,14,15]. Similar to other monoclonal antibodies such as omalizumab, which was used primarily in severe allergic asthma and then extended to chronic spontaneous urticaria, we may see changes in Benralizumab’s indication use also. Many autoinflammatory diseases were exacerbated by COVID-19 and monoclonal antibodies are a successful add-on treatment as seen in chronic spontaneous urticaria treated with omalizumab [25]. Thus, Benralizumab may also be an add-on treatment in selected severe cases, due to both efficacy and cost-benefit. 

To the best of our knowledge, this is the first report of Benralizumab and IVIg coadministration. The clinical rationale of IVIg therapy, in this case, was, on the one hand, the immunomodulatory properties of immunoglobulin administration in a possible autoimmune induced-vasculitis, and on the other hand, there was the context of a possible vasculitis associated with sepsis. Although cANCA and pANCA were negative in this case, the diagnostic value of anti-neutrophil cytoplasmic antibody testing is not absolute. Negative results do not rule out the possibility of cutaneous vasculitis with systemic features in the context of an autoimmune phenomenon or sepsis [26]. It is difficult to establish which treatment option, Benralizumab or IVIg, was more beneficial, but we can suspect that Benralizumab helped with the eosinophilic inflammation, while IVIg could have been effective for both sepsis and immunomodulation in a potential steroid and immunosuppressive- resistant autoimmune disease. The efficacy of IVIg in sepsis is due to the role of immunoglobulins in the recognition and clearance of pathogens and toxins, scavenging and inhibiting up- and downstream mediator gene transcription, and their anti-apoptotic effects on immune cells [27]. Although literature brings conflicting data on the topic, the use of intravenous immunoglobulins appears to be safe in sepsis and septic shock, as well as to be beneficial to sepsis-related inflammation and coagulopathy [27]. IgM-enriched formulations may be the most advantageous option in cases of sepsis [27]. We did not have this on hand, so we used a normal IVIg formulation which was available in our hospital’s pharmacy. In addition, IVIg contains anti-idiotypic antibodies, which explains their immunomodulatory action [28]. Thus, IVIg seemed to be a safe and rational choice for our patient.

In terms of eosinophils’ role in the immunopathology of the first case, it is difficult to state that their level increased in the context of the post-COVID illness, due to the fungus/bacterial infections, that it was triggered by the antibiotic or the fungal infections, or that it may have been due to a combination of two or more of them. Since the clinical status of the patient worsened abruptly and was associated with peripheral eosinophilia despite pulse therapy, we decided to “attack” the eosinophils/IL-5 axis with Benralizumab as well as to administer immunomodulatory IVIg and to continue broad-spectrum antibiotics and systemic antifungal agents.

Other diseases can mimic vasculitis and need to be considered. Apart from infection, several rare conditions including thrombotic disorders (antiphospholipid antibody syndrome, thrombotic thrombocytopenic purpura, sickle cell disease, embolization from atrial myxoma, and cholesterol emboli from atheroma), non-inflammatory vessel wall disorders (fibromuscular dysplasia, amyloidosis, scurvy, and vasospasm due to ergot) could mimic the presentation of vasculitis by causing ischemic manifestations or systemic symptoms. The patient was indeed known with thrombophilia (heterozygous factor V Leiden mutation and heterozygous MTHFR 1298C mutation), thrombosis of the left internal jugular, subclavian, and brachiocephalic vein; however, she was effectively anticoagulated. Aside from this, an infection could have been the main trigger of the vasculitis manifestations, and post-COVID-19 illness could equally be suspected. In the authors’ experience, many cases have presented with urticarial vasculitis in the post-COVID-19 phase, with both clinical and histopathological features compatible with this diagnosis.

With regard to the second case, steroids were continued after Benralizumab administration, due to the known risk of DRESS relapse after discontinuation of treatment. It is difficult to establish the evolution of the case in the absence of steroid use. However, we used a lower dose of corticosteroids for a shorter period of time, with no relapse of the disease. Thus, we might suspect that Benralizumab was beneficial in terms of possibly shortening the course of the disease as well as lowering the need for steroid therapy, subsequently leading to a reduction in the adverse effects of treatment.

In our experience, off-label Benralizumab successfully controlled progressive post-COVID-19 vasculitis despite conventional immunosuppressive treatment, as well as severe DRESS. In addition, to the best of our knowledge, this is the first report of the association of Benralizumab with high doses of immunoglobulin intravenous infusion as immunomodulatory treatment in eosinophilic vasculitis. Treatment induced the complete remission of symptoms and laboratory parameters and patients were released from the hospital symptom-free.

## 5. Conclusions

Off-label drug use remains an important public health issue, particularly for individuals with rare conditions or diseases not responsive to conventional treatments. We here report the successful administration of Benralizumab in severe necrotizing vasculitis during COVID-19 disease and in severe DRESS due to allopurinol, which did not respond to current treatment options.

## Figures and Tables

**Figure 1 jcm-11-06642-f001:**
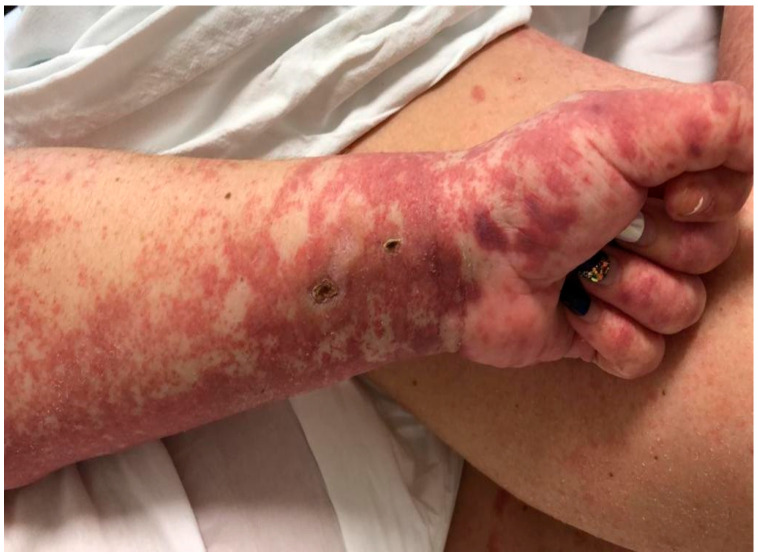
Vasculitic lesions on hands and forearms.

**Figure 2 jcm-11-06642-f002:**
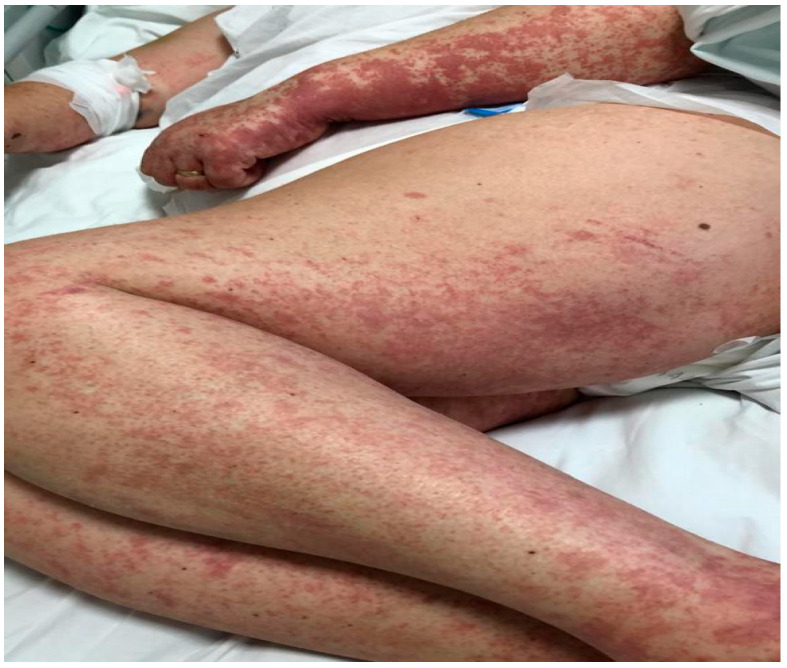
Vasculitic lesions on upper and lower limbs, with a tendency for generalization.

**Figure 3 jcm-11-06642-f003:**
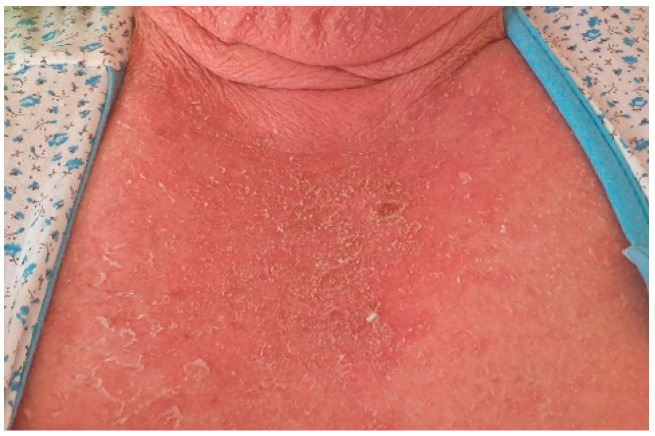
Erythematous skin rash with lamellar desquamation.

**Figure 4 jcm-11-06642-f004:**
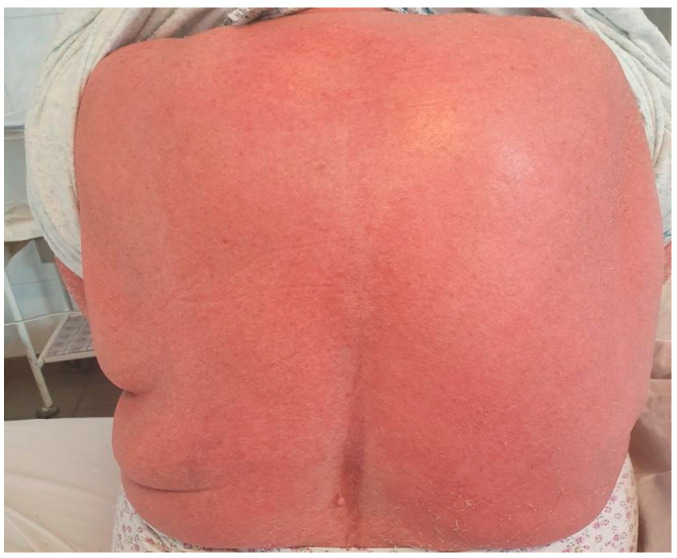
Erythematous skin rash with lamellar desquamation.

**Figure 5 jcm-11-06642-f005:**
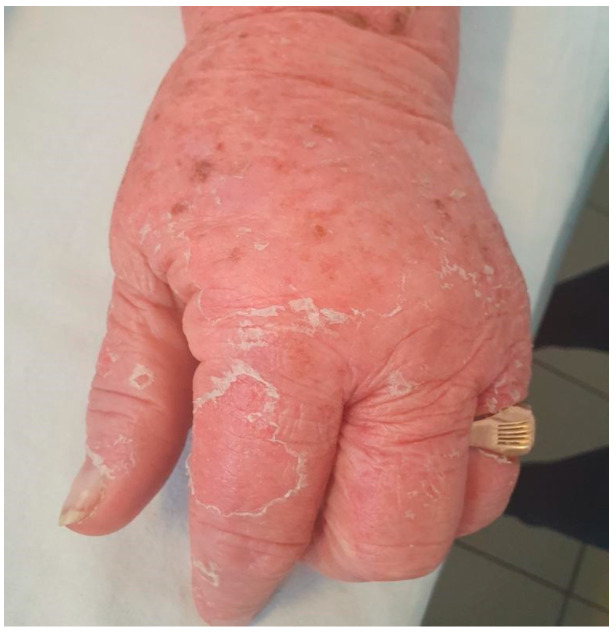
Erythematous skin rash with lamellar desquamation.

**Table 1 jcm-11-06642-t001:** Laboratory parameters before and after administration of Benralizumab.

Variables (n.r.) (Unit of Measurement)	Days before and after the Administration						
	−3	−2	−1	0	1	2	3
Leukocytes (4–10) (10^3^/μL)	12.32	11.93	13.71	9.46	8.60	9.40	9.40
Neutrophils (30–75) (%)	68.9	68.8	70	80.6	66.9	56.1	58.1
Neutrophils (2–7.5) (10^3^/μL)	8.49	8.21	9.61	7.62	5.76	5.27	7.47
Eosinophils (1–4) (%)	3.9	3.9	4.7	1.9	0.0	0.00	0.00
Eosinophils (0.05–0.4) (10^3^/μL)	0.48	0.46	0.64	0.18	0.00	0.00	0.00
Lymphocytes (20–40) (%)	18.2	19.1	18.5	14.9	27.8	36.7	32.9
Lymphocytes (1.5–4) (10^3^/μL)	2.24	2.28	2.53	1.41	2.39	3.45	4.23
PLT (140–440) (10^3^/μL)	437	459	198	543	618	614	588
CRP (0–1) (mg/dL)	9.7	9.37	12.37	10.44	6.22	3.28	2.05
D-dimers (<0.5) (ng/mL)	6.24	6.24	6.73	7.76	6.22	5.19	3.1

n.r.: normal range.

**Table 2 jcm-11-06642-t002:** RegiSCAR scoring for drug reaction with eosinophilia and systemic symptom syndrome.

Parameter	Score			Comments
	−1	0	1	
Fever ≥ 38.5 °C	No/unknown	Yes		
Lymphadenopathy		No/Unknown	Yes	>1 cm, at least two sites
Eosinophilia ≥ 0.7 × 109 or ≥10% if leucopenia		No/Unknown	Yes	Score 2 points of ≥1.5 × 109
Atypical lymphocytes		No/Unknown	Yes	
Skin rash				Suggestive features: ≥2 facial edemas, purpura, infiltration, and desquamation
Rash suggestive of DRESS	No	Unknown	Yes	
Extent ≥ 50% of BSA		No/Unknown	Yes	
Skin biopsy suggestive of DRESS	No	Yes/Unknown		
Organ involvement		No	Yes	1 point for each organ involvement, maximum score: 2
Disease duration ≥ 15 days	No/unknown	Yes		
Exclusion of other causes		No/Unknown	Yes	1 point if three of the following tests are performed and are negative: HAV, HBV, HCV, mycoplasma, chlamydia, ANA, and blood culture

**Table 3 jcm-11-06642-t003:** Laboratory parameters before and after the administration of Benralizumab.

Variables (n.r.) (Unit of Measurement)	Days before and after the Administration				
	−1	0	1	2	3
Leukocytes (4–10) (10^3^/μL)	12.4	12.5	9.99	9.42	9.5
Neutrophils (30–75) (%)	54	55.1	87.1	87.6	87.3
Neutrophils (2–7.5) (10^3^/μL)	6.70	6.71	8.70	8.25	8.6
Eosinophils (1–4) (%)	25	21.6	0.1	0.9	0.00
Eosinophils (0.05–0.4) (10^3^/μL)	2.59	2.62	0.01	0.09	0.00
Lymphocytes (20–40) (%)	12.5	12	9.0	7.2	7.12
Lymphocytes (1.5–4) (10^3^/μL)	1.45	1.45	0.90	0.68	0.69
PLT (140–440) (10^3^/μL)	532	532	533	484	440
CRP (0–1) (mg/dL)	1.59	1.59	1.6	1.58	1.29
Urea nitrogen (18–48) (mg/dL)	180	183	166	157	155
Creatinine (0–1.10) (mg/dL)	1.67	1.66	1.58	1.53	1.54

n.r.: normal range.

## Data Availability

Data are available at the Allergology Department, Octavian Fodor Institute of Gastroenterology and Hepatology, Cluj Napoca.

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
