# Peer review of "Off-Label Benralizumab in Severe Non-Necrotizing Eosinophilic Vasculitis following Critical COVID-19 Disease and in DRESS"

_jcm, 2022, doi:10.3390/jcm11226642_

Round 1

Reviewer 1 Report

This is an interesting manuscript describing 2 cases of eosinophil mediated diseases relatively rare with the success of single dose Benralizumab.

I have few questions/remarks

- in the first case why there were "rare eosinophils" on Biopsy yet you  decided to treat with "anti-eosinophils treatment"? is it d/t the corticosteroids the patient received before the  biopsy? it should be explained in the manuscript .

-in the 1 case-  the authors should describe shortly what is the efficacy of IVIG in similar cases and whether they assume that maybe the IVIG was beneficial to the patient rather then Benralizumab?  

Author Response

Response Reviewer 1

Dear Reviewer,

The authors of the communication “Off-label Benralizumab in severe non-necrotizing eosinophilic vasculitis following critical COVID-19 disease and in DRESS” express deep gratitude to you for agreeing to review our work. Thank you very much for the professional analysis of our manuscript and for the remarks made. The authors have made the necessary changes to the text in accordance with the comments made by reviewers.

This is an interesting manuscript describing 2 cases of eosinophil mediated diseases relatively rare with the success of single dose Benralizumab.

I have few questions/remarks

- in the first case why there were "rare eosinophils" on Biopsy yet you  decided to treat with "anti-eosinophils treatment"? is it d/t the corticosteroids the patient received before the  biopsy? it should be explained in the manuscript .

Response: Thank you for your valuable comment!

The patient received pulse therapy with methylprednisolone (intravenous 250 mg, for 3 days, with day one being the first day of vasculitic skin rash onset). We decided to start pulse therapy because of the rapid progression of the skin rash, along with the worsening of the clinical status. A skin biopsy was done on the second day of pulse therapy. However, despite high doses of corticosteroids, the skin rash progressed and peripheral eosinophilia developed, which prompted the idea of Benralizumab administration, along with IVIg. Since initially the patient had eosinophils within normal ranges, we consider that eosinophilia could be placed in the context of the post COVID illness and the vasculitic picture (initially, eosinophils were 480/mmc and increased to 640/mmc).

Thus, we added in the text:

“The purpuric rash was initially localized over hands and legs, and consisted in round, 1-3 mm in diameter skin lesions with a rapid tendency to coalesce and form plaques, up to the point that within the following 12 hours after onset, it was affecting more than 60% of the BSA. Given the rapid progression of the skin rash along with the worsening of the clinical status despite broad-spectrum intravenous antibiotics, pulse therapy with methylprednisolone is initiated (intravenous injection, 250 mg/day−1 for 3 days, with day 31 of admission being the first day of corticosteroid administration). Skin biopsy, which was done on day 32 of admission and the second day of pulse therapy, revealed non-necrotizing small vessel vasculitis with lymphoplasmacytic infiltrates, and rare eosinophils. In addition to the rapid progression of vasculitic lesions, the patient develops slight peripheral eosinophilia despite methylprednisolone administration.”

-in the 1 case-  the authors should describe shortly what is the efficacy of IVIG in similar cases and whether they assume that maybe the IVIG was beneficial to the patient rather then Benralizumab?  

Response:Thank you for your valuable comment!

To our knowledge, this is the first report of Benralizumab and IVIg coadministration. The clinical rationale of IVIg therapy, in this case, was, on the one hand, the immuno-modulatory properties of immunoglobulin administration in a possible autoimmune induced-vasculitis, and on the other hand, there was the context of a possible vasculitis as-sociated with sepsis. Although cANCA and pANCA were negative in this case, the diagnostic value of anti-neutrophil cytoplasmic antibody testing is not absolute.Negative results do not rule out the possibility of cutaneous vasculitis with systemic features in the context of an autoimmune phenomenon or sepsis. It is difficult to establish which treatment option, Benralizumab or IVIg, was more beneficial, but we can suspect that Benralizumab helped with the eosinophilic inflammation, while IVIg could have been effective for both sepsis and immunomodulation in a potential steroid and immunosuppressive- resistant autoimmune disease. The efficacy of IVIg in sepsis is due to the role of immunoglobulins in the recognition and clearance of pathogens and toxins, scavenging and inhibition of up- and downstream mediator gene transcription, and antiapoptotic effects on immune cells. Although literature brings conflicting data on the topic, the use of intravenous immunoglobulins appears to be safe in sepsis and septic shock, as well as to be beneficial to sepsis-related inflammation and coagulopathy. IgM-enriched formulations may be the most advantageous option in cases of sepsis.  We did not have this on hand, so we used a normal IVIg formulation which was available in our hospital’s pharmacy. In addition, IVIg contains anti-idiotypic antibodies, which explains their immunomodulatory action. Thus, IVIg seemed to be a safe and rational choice for our patient.

This was added in the manuscript as well, in the Discussions section (page 6 paragraph 1), with the corresponding references.

Reviewer 2 Report

The authors reported interesting cases in which Benralizumab was effective on two rare diseases; one was post-COVID-19 vasculitis, and the other was DRESS. Particularly the former might be the first case. This report will be helpful information for clinicians. However, there are several concerns about this manuscript.

1. I am trying to understand why the authors used Benralizumab for case 1, in which eosinophilic inflammation was not apparent. It is necessary to show the reasons for the use of Benralizumab.

2. The authors diagnosed case 1 as vasculitis based on pathological findings. Did the author find the destruction of the vessel wall structure? The authors should provide an image of pathology for diagnosing vasculitis in case1.

3. I am not sure that all the skin findings can be explained by vasculitis alone in case1.

4. It is well known that COVID-19 infection causes the autoimmune phenomenon.

Did the authors examine autoantibodies such as ANA, anti-phospholipid antibodies, and ANCA?  Did the authors think immunological abnormalities causing vascular lesions was due to COVID-19 infection or due to bacterial/ fungus infection?

5. Case 1 was an exciting one. Benralizumab improved skin lesions in Case1, indicating eosinophils and IL-5 axis play an important role in developing skin/ vascular lesions, although eosinophilic inflammation was not apparent in this case. It is better to discuss the role of eosinophils/ IL-5 axis in the development of post-COVID vasculitis and this case.

Are there any reports about eosinophils or IL-5 in post-COVID illness?

6. In DRESS, an immune abnormality is driving eosinophilic inflammation. Administration of Benralizumab may temporarily suppress eosinophilic inflammation but not correct the immune abnormality. Was there a relapse of symptoms of DRESS when the effect of Benralizumab was diminished?

7. Words “ off-label use” was necessary?

8Did the authors obtain consent for publication from the patient?

Author Response

Response Reviewer 1

Dear Reviewer,

The authors of the communication “Off-label Benralizumab in severe non-necrotizing eosinophilic vasculitis following critical COVID-19 disease and in DRESS” express deep gratitude to you for agreeing to review our work. Thank you very much for the professional analysis of our manuscript and for the remarks made. The authors have made the necessary changes to the text in accordance with the comments made by reviewers.

The authors reported interesting cases in which Benralizumab was effective on two rare diseases; one was post-COVID-19 vasculitis, and the other was DRESS. Particularly the former might be the first case. This report will be helpful information for clinicians. However, there are several concerns about this manuscript.

  1. I am trying to understand why the authors used Benralizumab for case 1, in which eosinophilic inflammation was not apparent. It is necessary to show the reasons for the use of Benralizumab.

Answer: We decided in favor of Benralizumab administration because, despite high doses of corticosteroids, the skin rash progressed rapidly, with 60% of the BSA being affected within the first day after the onset of purpuric skin lesions, followed by an increase in peripheral eosinophilia. Despite intravenous broad-spectrum antibiotics and pulse therapy, patient’s clinical status worsened, with skin rash compatible with vasculitis and mild peripheral eosinophilia as key features. Thus, we suspected a potential autoimmune reaction with eosinophilic inflammation following severe COVID-19 disease or vasculitis in the context of sepsis, which prompted the idea of the compassionate use of Benralizumab along with IVIg.

To further explain this in the text, we added the following data to the case description:

“The purpuric rash was initially localized over hands and legs, and consisted in round, 1-3 mm in diameter skin lesions with a rapid tendency to coalesce and form plaques, up to the point that within the following 12 hours after onset, it was affecting more than 60% of the BSA. Given the rapid progression of the skin rash along with the worsening of the clinical status despite broad-spectrum intravenous antibiotics, pulse therapy with methylprednisolone is initiated (intravenous injection, 250 mg/day−1 for 3 days, with day 31 of admission being the first day of corticosteroid administration). Skin biopsy, which was done on day 32 of admission and the second day of pulse therapy, revealed non-necrotizing small vessel vasculitis with lymphoplasmacytic infiltrates, and rare eosinophils. In addition to the rapid progression of vasculitic lesions, the patient develops slight peripheral eosinophilia despite methylprednisolone administration.”

  1. The authors diagnosed case 1 as vasculitis based on pathological findings. Did the author find the destruction of the vessel wall structure? The authors should provide an image of pathology for diagnosing vasculitis in case1.

Answer: The destruction of vessel wall structure was not mentioned in the histopathology report. Unfortunately, we do not have an image from the Pathology Department; we only have the result that has been given by the Anatomopathology doctors.

  1. I am not sure that all the skin findings can be explained by vasculitis alone in case1.

Answer: Other diseases can mimic vasculitis and need to be considered. Apart from infection, several rare conditions including thrombotic disorders (antiphospholipid antibody syndrome, thrombotic thrombocytopenic purpura, sickle cell disease, embolization from atrial myxoma, cholesterol emboli from atheroma), non‐inflammatory vessel wall disorders (fibromuscular dysplasia, amyloidosis, scurvy, vasospasm due to ergot) could mimic the presentation of vasculitis by causing ischaemic manifestations or systemic symptoms. The patient was indeed known with thrombophilia (heterozygous factor V Leiden mutation and heterozygous MTHFR 1298C mutation), thrombosis of the left internal jugular, subclavian, and brachiocephalic vein; however, she was effectively anticoagulated. Aside from this, an infection could have been the main trigger of the vasculitis manifestations, and post-COVID-19 illness could equally be suspected. In the authors’ experience, many cases have presented with urticarial vasculitis in the post-COVID-19 phase, with both clinical and histopathological features compatible with this diagnosis.

This was added to the “Discussion” section of the manuscript page 6 paragraph 2.

  1. It is well known that COVID-19 infection causes the autoimmune phenomenon.

Did the authors examine autoantibodies such as ANA, anti-phospholipid antibodies, and ANCA?  Did the authors think immunological abnormalities causing vascular lesions was due to COVID-19 infection or due to bacterial/ fungus infection?

Answer: ANA, anti-phospholipid antibodies, and ANCA were tested and negative results were obtained (this was added to the case description following your observation). However, we acknowledge that the negative predictive value is not absolute in cases of autoimmune phenomenon. Thus, the following was added to the section “Discussion” page 6 paragraph 3:

“Although cANCA and pANCA were negative in this case, the diagnostic value of anti-neutrophil cytoplasmic antibody testing is not absolute. Negative results do not rule out the possibility of cutaneous vasculitis with systemic features in the context of an autoimmune phenomenon or sepsis.” 

Correspondent references were added to the text.

We considered both post-COVID-19 vasculitis and an infectious associated vasculitis. This was further explained in the text, placed in the context of treatment choices, in the “Discussion” section page 6 1st paragraph and corresponding references were also added:

“To our knowledge, this is the first report of Benralizumab and IVIg coadministration. The clinical rationale of IVIg therapy, in this case, was, on the one hand, the immuno-modulatory properties of immunoglobulin administration in a possible autoimmune induced-vasculitis, and on the other hand, there was the context of a possible vasculitis as-sociated with sepsis. Although cANCA and pANCA were negative in this case, the diagnostic value of anti-neutrophil cytoplasmic antibody testing is not absolute. Negative results do not rule out the possibility of cutaneous vasculitis with systemic features in the context of an autoimmune phenomenon or sepsis. It is difficult to establish which treatment option, Benralizumab or IVIg, was more beneficial, but we can suspect that Benralizumab helped with the eosinophilic inflammation, while IVIg could have been effective for both sepsis and immunomodulation in a potential steroid and immunosuppressive- resistant autoimmune disease. The efficacy of IVIg in sepsis is due to the role of immunoglobulins in the recognition and clearance of pathogens and toxins, scavenging and inhibition of up- and downstream mediator gene transcription, and antiapoptotic effects on immune cells. Although literature brings conflicting data on the topic, the use of intravenous immunoglobulins appears to be safe in sepsis and septic shock, as well as to be beneficial to sepsis-related inflammation and coagulopathy. IgM-enriched formulations may be the most advantageous option in cases of sepsis.  We did not have this on hand, so we used a normal IVIg formulation which was available in our hospital’s pharmacy. In addition, IVIg contains anti-idiotypic antibodies, which explains their immunomodulatory action. Thus, IVIg seemed to be a safe and rational choice for our patient.”

  1. Case 1 was an exciting one. Benralizumab improved skin lesions in Case1, indicating eosinophils and IL-5 axis play an important role in developing skin/ vascular lesions, although eosinophilic inflammation was not apparent in this case. It is better to discuss the role of eosinophils/ IL-5 axis in the development of post-COVID vasculitis and this case.

Are there any reports about eosinophils or IL-5 in post-COVID illness?

Answer: The following paragraphs were added to the “Discussion” section in response to your comment page 6 paragraphs 2 and 3:

“Eosinophils play an important role in protecting the host against viral infections, and several publications discuss the eosinophils/IL-5 axis in COVID-19 illness. SARS-CoV-2 infection stimulates a complex activation of the immune response. In the first phase of the infection, the participation of eosinophils is probably appropriate and beneficial, by limiting the viral replication, while in the second phase of clinical pulmonary symptoms and the third phase of the immunopathological response, which results in a cytokine storm, they may take part in a maladaptive immune response and immunopathology. Mounting evidence shows that COVID-19 disease exhibits peculiar features in terms of immune response. Eosinopenia has been indicated for the prediction of poor prognosis in COVID-19 disease.  The increase in eosinophils over the course of the disease from initially low levels could be a positive indicator of clinical improvement. By contrast, it has been indicated that current studies do not support the use of eosinopenia for the diagnosis of COVID-19. It has also been suggested that eosinopenia may not be associated with an unfavorable progression of COVID-19 and that eosinopenia may show more diagnostic—and, eventually, prognostic—value than real participation in COVID-19’s pathology. Therefore, the role of eosinophils in COVID-19 remains controversial. A study published in 2021 on the behavior of eosinophil counts in recovered and deceased COVID-19 patients over the course of the disease shows that the increase in eosinophils in the recovered patients protected against the exacerbation of inflammation, protection that was lacking in the deceased patients.”

“In terms of eosinophils’ role in the immunopathology of the first case, it is difficult to state that their level increased in the context of the post-COVID illness, due to the fungus/bacterial infections or that it was triggered by the antibiotic or the fungal infections or maybe a combination of two or more of them. Since the clinical status of the patient worsened abruptly and was associated with peripheral eosinophilia despite pulse-therapy, we decided to “attack” the eosinophils/IL-5 axis with Benralizumab as well as to administer immunomodulatory IVIgs and to continue broad spectrum antibiotics and systemic antifungal agents. ”

  1. In DRESS, an immune abnormality is driving eosinophilic inflammation. Administration of Benralizumab may temporarily suppress eosinophilic inflammation but not correct the immune abnormality. Was there a relapse of symptoms of DRESS when the effect of Benralizumab was diminished?

Answer: We completed the treatment given after Benralizumab administration. In response to your question, the following was added to the case description text:

“Since DRESS can relapse after treatment discontinuation, we discussed the necessity of long-term corticotherapy with the patient. A gradual dose reduction of prednisone regimen was agreed upon and given over 5 weeks (60 mg/d x 5 days, 40 mg/d x 5 days, 20 mg/d x 5 days, 10 mg/d x 5 days, 5 mg/d x 5 days, followed by 2,5 mg/d x 5 days, and then 2,5 mg/d on alternate days for 6 days) so as to prevent rapid reconstitution of specific immune responses. The evolution of the case was favorable, both clinically and in terms of lab tests’ results. DRESS did not relapse after wearing off Benralizumab, more likely due to the benefits of long term corticosteroid administration.”

Also, the following was added to the “Discussion” section:

“With regards to the second case, steroids were continued after Benralizumab administration, due to the known risk of DRESS relapse after discontinuation of treatment. It is difficult to establish the evolution of the case in the absence of steroid use. However, we used a lower dose of corticosteroids for a shorter period of time, with no relapse of the disease. Thus, we might suspect that Benralizuman was beneficial in terms of possibly shortening the course of the disease as well as lowering the need for steroid therapy, subsequently leading to a reduction of adverse effects of treatment.”

  1. Words “ off-label use” was necessary?

Answer: Benralizumab was used for conditions that are different from its approved indications, so we consider that “off-label” is correct. However, the association of words is unfortunate, so we erased the word “use” from the title and body of the manuscript. 

8 Did the authors obtain consent for publication from the patient?

Answer: I hereby confirm that the authors have obtained consent for publication from the patients, as we mentioned in the manuscript.

Thank you very much for your valuable comments!

Round 2

Reviewer 2 Report

The manuscript has been improved.